# Decreased expression of Ly-1 antibody reactive clone (Lyar) triggers enhanced adipogenesis of bone marrow mesenchymal stromal cells in aged bone marrow

Yu Shinyashiki[1]☯, Yuta Onodera[2]☯, Yusuke Kawashima[3], Natsumi Iwawaki[2], Toshiyuki Takehara[2], Koji Goto[1], Takeshi Teramura[2]*

1 Department of Orthopedic Surgery, Kindai University Faculty of Medicine, Osaka, Japan, 2 Division of Cell Biology for Regenerative Medicine, Institute of Advanced Clinical Medicine, Kindai University Faculty of Medicine, Osaka, Japan, 3 Department of Applied Genomics, Kazusa DNA Research Institute, Chiba, Japan

☯ These authors contributed equally to this work.
* teramura@med.kindai.ac.jp

## Abstract

Accumulation of adipocytes within the bone marrow is a frequently observed during aging. However, the molecular mechanisms underlying aberrant adipocyte differentiation in aged bone marrow remain largely unclear. In this study, we identified Ly-1 antibody reactive clone (Lyar) as an interacting partner of TGF-β activated kinase 1 (Tak1), a key molecule of non-canonical TGF-β signaling, through a proteomics approach, and demonstrated its involvement in the regulation of aging-related enhancement of adipogenesis. Lyar was not only implicated in the regulation of BMMSC proliferation but also may partly mediate the inhibitory effects of Bromodomain-containing protein 2 (Brd2). An age-associated decline in Lyar expression was associated with a reduction in FGF2–PI3K–Akt1 signaling activity in aged bone marrow. These findings suggest that Lyar may act as a context-dependent modulator of TGF-β signaling and may be involved in regulating proliferation and differentiation in BMMSCs. The age-related loss of Lyar may contribute to the complex mechanisms underlying enhanced adipogenesis in aged bone marrow, providing new insights into the regulation of mesenchymal stem cell fate during aging.

## Introduction

The number and size of adipocytes in the bone-marrow increase along with aging [1,2]. Histological analyses of bone biopsies from elderly and osteoporotic patients have revealed an inverse relationship between bone mass and marrow adipose tissue (MAT) accumulation, suggesting a potential pathophysiological link [3]. Recent studies further support that elevated MAT levels are associated with osteoporosis and an increased risk of bone fractures [4,5]. Moreover, an excessive presence

**Data availability statement:** All relevant data are within the manuscript and its Supporting information files.

**Funding:** The author(s) received no specific funding for this work.

**Competing interests:** The authors have declared that no competing interests exist.

of adipocytes has been shown to impair hematopoietic reconstitution and fracture healing, due to disruption of the osteogenic and hematopoietic niches within the long bones [6].

The age-associated increase in marrow adiposity has primarily been attributed to an imbalance between adipogenic and osteogenic differentiation [7]. Both adipocytes and osteoblasts originate from bone marrow mesenchymal stem cells (BMMSCs). However, the cellular and molecular mechanisms driving age-related alterations in BMMSCs and their contribution to MAT formation remain largely unclear.

During aging, multiple signaling pathways become dysregulated, exemplified by senescence-associated secretory phenotype (SASP) factors secreted by senescent cells. Among these, elevated levels of transforming growth factor-beta (TGF-β) represent one of the most robust and consistently observed features of aged tissues. TGF-β exerts complex and context-dependent effects on stem cell proliferation and differentiation via both SMAD-dependent canonical and SMAD-independent non-canonical signaling pathways. Recent studies have underscored the pivotal role of TGF-β–activated kinase 1 (Tak1), a key mediator of non-canonical TGF-β signaling, in regulating the self-renewal capacity of muscle satellite cells [8], hematopoietic stem cells [9], and MSCs [10]. However, despite persistently elevated TGF-β levels in aged tissues, the mechanisms underlying the paradoxical decline in stem cell populations and the shift toward adipocyte differentiation remain poorly understood.

In this study, we identify Lyar as a novel regulatory molecule that interacts with Tak1 and may contribute to the regulation of proliferation and differentiation in murine BMMSCs. We show that Lyar expression is significantly downregulated in BMMSCs from aged mice, suggesting that Lyar may act as a potential contributor within the TGF-β–Tak1 signaling axis and may be associated with age-related impairment in MSC self-renewal and lineage specification.

## Materials and methods

### Ethical consideration for animal studies

All procedures related to animal handling, care, surgery, and sacrifice were approved by the Institutional Animal Care and Use Committee of Kindai University as the approved research project number KAME2022−062 and performed in accordance with institutional guidelines and regulations. All methods are reported in accordance with ARRIVE guidelines.

### Preparation of mouse bone marrow mesenchymal stromal cells (BMMSCs)

All mice used in this study were purchased from CLEA Japan, Inc (Osaka, Japan). The mice were bred and maintained under controlled conditions in accordance with institutional and international guidelines for the care and use of laboratory animals. Three to four-week-old male C57BL/6J (B6) mice were used as young model, and 20- to 24-month-old male B6 mice were used as aged model. Before sacrifice, mice were euthanized by isoflurane overdose in an anesthesia chamber, followed by cervical dislocation. Mouse BMMSCs were isolated as previously described. In brief,

cleaned femurs and tibias were cut into small pieces and treated with 0.1% collagenase type II (Wako, Tokyo, Japan) for 15 minutes. Single-cell suspensions containing BMMSCs were collected, washed twice with PBS, and plated on cell culture dishes (Sumilon, Sumitomo Bakelite Co. Ltd., Tokyo, Japan) in αMEM (Thermo Fisher Scientific, Waltham, MA, USA) supplemented with 200 mM L-glutamine and 10% fetal bovine serum (CORNING, Lowell, MA, USA) under 5% $CO_2$ and 5% $O_2$ at 37°C. The following day, the medium was replaced to remove dead cells and debris. After 10 days of culture, BMMSCs were detached using TrypLE (Thermo Fisher Scientific) for further experiments. BMMSCs within three passages were used in the experiment.

For adipogenic differentiation, BMMSCs were cultured in DMEM containing 10% FCS and supplemented with 0.1 mM indomethacin, 0.45 mM IBMX, 10 µg/mL insulin, and 1 µM dexamethasone for 14 days. For osteogenic differentiation, BMMSCs were cultured in DMEM containing 10% FCS and supplemented with 100 nM dexamethasone, 50 mM ascorbic acid, 10 mM β-glycerophospahte for 14 days.

## Quantitative RT-PCR (qRT-PCR)

Total RNA was isolated using TRI Reagent® (Molecular Research Center, Inc., Cincinnati, OH, USA). One hundred nanograms of total RNA were reverse-transcribed into cDNA using the PrimeScript® RT Master Mix Kit (TAKARA Bio Inc., Shiga, Japan). Quantitative PCR was then performed using gene-specific primers and the SYBR Green method with Perfect Real Time SYBR® Green II (Takara Bio Inc.). PCR amplification was carried out on a Thermal Cycler Dice® Real Time System Single under the following conditions: 95°C for 20 s, followed by 40 cycles of 95°C for 5 s and 60°C for 30 s. Relative gene expression levels were quantified by normalizing Ct (threshold cycle) values to Gapdh ($\Delta Ct = Ct\_target - Ct\_GAPDH$) and comparing them with a calibrator using the ΔΔCt method ($\Delta\Delta Ct = \Delta Ct\_sample - \Delta Ct\_control$). Primer sequences are listed in S1 Table.

## Western blot (WB) analysis

Samples were homogenized in SDS buffer (4% SDS, 125 mM Tris–glycine, 10% b- mercaptoethanol, 2% bromophenol blue in 30% glycerol) and centrifuged at 10,000 rcf at 4°C for 10 min to remove debris. Aliquots were subjected to sodium dodecyl sulphate- polyacrylamide gel electrophoresis followed by electrotransfer onto a PVDF membrane (Hybond-P; Cytiva, Tokyo, Japan). The blotted membranes were blocked for 1hr with Block Ace (Dainippon Sumitomo Pharma, Osaka, Japan) or for 10 min with 0.1% PVA-TBS and then probed with primary antibody overnight at 4°C. Detection was conducted with horseradish peroxidase (HRP)-conjugated secondary antibodies (all antibodies were purchased from CST) and either the ECL prime Western blotting detection system (Cytiva) or Immunostar® LD (Wako). The immunolabelled membranes were analysed using a CCD- based chemiluminescence analyser (Amersham™ Imager 680; Cytiva).

## Treatment of primary MSCs with siRNA

Primary MSCs from mice bone marrow tissues were transfected with miRNA mimic or siRNA using Lipofectamine® RNAi-MAX (Thermo Fisher Scientific) following the manufacturer's instructions. The siRNA blocking Lyar were synthesized with the following sequences (Japan Bio Services Co.,LTD., Saitama, Japan):

sense sequence- CCA UUA AGG CUG UUU UGA AdTdT and antisense sequence- UUC AAA ACA GCC UUA AUG GdTdC. Briefly, primary MSCs were plated at $5 \times 10^4$ cells per well of 24-well multiplate and transfected at 85% confluency. The final siRNA concentration was 20nM, and siRNA was complexed with Lipofectamine® RNAiMAX at a 1:3 ratio, according to the manufacturer's protocol. A non-targeting negative control siRNA (Japan Bio Services Co.,LTD., Saitama, Japa) was used in parallel. Forty-eight hours after transfection, cells were collected for RT-PCR, WB or proteomics.

## Immunoprecipitation (IP)

Cells were lysed in ice-cold lysis buffer composed with 50 mM Tris-HCl pH 7.5, 150 mM NaCl, 1% NP-40, 1 mM EDTA on ice for 30 minutes. Cell debris was removed by centrifugation at 12,000×g for 15 minutes at 4°C. After centrifugation, supernatants were quantified by BCA assay. For each IP, 500 μg of total protein in a final volume of 500 μL was used. The supernatant was collected and quantified using a BCA protein assay kit (Thermo Fisher Scientific). To reduce nonspecific binding, lysates were pre-cleared with protein G-conjugated beads (Dynabeads Protein G, Thermo Fisher Scientific) for 1 hour at 4°C with rotation, followed by magnetic separation or centrifugation. The cleared lysates were incubated with 1 μg primary antibody overnight at 4°C with gentle rotation. Subsequently, 50 μL of protein G beads were added and incubated for an additional 2 hours at 4°C. Beads were washed 3 times with lysis buffer to remove nonspecifically bound proteins.

The bound proteins were eluted by boiling the beads in SDS-PAGE sample buffer at 95 °C for 5 minutes. The eluates were subjected to SDS-PAGE and immunoblotting using appropriate primary and secondary antibodies (S2 Table).

## On-bead digestion of IP samples for proteome analysis

500 ng of trypsin/Lys-C Mix (CAT# V5072, Promega, Madison, WI, USA) were added to the IP-beads soaked in 50 mM ammonium bicarbonate, and then mixed gently at 37 °C for 18 hours to digest proteins. The digested peptides were trans-ferred to a new 1.5 mL tube. The collected sample was treated with 10 mM dithiothreitol at 50 °C for 30 min and alkylated using 30 mM iodoacetamide at room temperature for 30 min while being protected from light. Subsequently, the alkylated sample was acidified with 5% trifluoroacetic acid (TFA) and desalted using an SDB-Stage tip (7820–11200, GL Sciences Inc., Tokyo, Japan) according to the manufacturer's protocol, followed by drying in a centrifugal evaporator (miVac Duo concentrator, Genevac Ltd., Ipswich, UK). The dried peptides were redissolving in 3% ACN containing 0.1% TFA.

## Generation of HA-tagged Tak1–constitutively expressing ES cells

C57BL/6 ES cells [11] were used and maintained under standard culture conditions. Full-length Tak1 cDNA fused with a HA tag was cloned into a piggyBac transposon vector and introduced by electroporation using a NEPA21 electroporator (Nepa Gene Co., Ltd., Tokyo, Japan). ES cells were seeded onto gelatin-coated dishes, and 48 h later, drug selection was initiated with 1 μg/mL puromycin (Thermo Fisher Scientific).

## Mass spectrometry of IP samples

3.5 μL of redissolved peptides was directly injected onto a 75 μm×20 cm PicoFrit emitter (New Objective, Woburn, MA, USA) packed in-house with C18 core-shell particles (CAPCELL CORE MP 2.7 μm, 160 Å material; Osaka Soda Co., Ltd., Osaka, Japan) at 60 °C and then separated with a 120-min gradient (A=0.1% formic acid in water, B=0.1% formic acid in 80% ACN) consisting of 8–30% B in 0–90 min, and 30–70% B in 90–120 min at a flow rate of 100 nL/min using an Ulti-Mate 3000 RSLCnano liquid chromatography system (Thermo Fisher Scientific). Peptides eluting from the column were analyzed on an Orbitrap Exploris 480 (Thermo Fisher Scientific) for overlapping window (data independent acquisition) DIA-MS [12,13]. MS1 spectra were collected in the range of 495–745 m/z at 30,000 resolution to set an automatic gain control target of 3e6 and maximum injection time of "auto". MS2 spectra were collected in the range of more than 200 m/z at 45,000 resolution to set an automatic gain control target of 3e6, maximum injection time of "auto", and stepped normalized collision energy of 22, 26, and 30%. An isolation width for MS2 was set to 4 m/z and overlapping window patterns in 500–740 m/z were used window placements optimized by Skyline. MS files were searched against the mouse UniProt reference proteome using Scaffold DIA. The Scaffold DIA search parameters were as follows: experimental data search enzyme, trypsin; maximum missed cleavage sites, 1; precursor mass tolerance, 8 ppm; fragment mass tolerance, 8 ppm; static modification, cysteine carbamidomethylation. The protein identification threshold was set both peptide and protein false discovery rates of less than 1%.

To explore molecules interacting with Lyar, BMMSCs transiently transfected to overexpress FLAG-Lyar for 48 h were analyzed using the same methods as described above.

## Shotgun mass spectrometry–based proteomic analysis of BMMSCs from aged and young mice

BMMSCs were harvested from the bone marrow of 2-year-old and 4-week-old C57BL/6 mice and maintained in primary culture. A total of $5 \times 10^6$ cells were subjected to analysis without passaging. The peptide mixtures were reconstituted in 0.1% formic acid and subjected to liquid chromatography using the Vanquish Neo UHPLC system (Thermo Fisher Scientific) equipped with an Aurora Ultimator column (IonOptics, Fitzroy, VIC, Australia). Mass spectrometric analysis was performed on an Orbitrap Exploris 480 (Thermo Fisher Scientific). Mass spectrometry data were processed using Thermo Proteome Discoverer ver 2.5.0.400. The raw files were searched against the UniProt Human database using the SEQUEST algorithm. Peptide and protein identifications were filtered for a false discovery rate (FDR) of <1% at both the peptide and protein levels. Label-Free Quantification (LFQ) was performed to assess the relative abundance of proteins across samples.

## Statistical analysis

All experiments were analyzed using data obtained from three independent biological replicates. Statistical analyses were performed using JMP version 18 (SAS Institute Inc., Cary, NC, USA). Multiple comparisons were conducted using Tukey's honestly significant difference (HSD) test. A p value $< 0.05$ was considered statistically significant.

## Results

### Bone marrow tissues from aged mice exhibit activation of non-canonical TGFβ signaling and a shift toward adipogenic differentiation

To assess whether non-canonical TGFβ1 signaling is altered in aged bone marrow, we harvested bone marrow tissue from young (3-week-old) and aged (2-year-old) mice. Quantitative RT-PCR analysis revealed that *Tgfb1* mRNA levels were approximately 10-fold higher in aged bone marrow compared to young controls, consistent with previous findings (Fig 1A). Immunoblot analysis further confirmed a marked elevation in Tgfβ1 protein levels in aged samples. Notably, phosphorylation of Tak1 was substantially increased in aged bone marrow (Fig 1B). Concurrently, expression levels of adipogenic marker genes *Cebpa*, *Fabp4*, and *Pparg* were significantly upregulated at both the mRNA and protein levels, indicating a pronounced shift toward adipogenic differentiation in the aged bone marrow microenvironment (Fig 1C and 1D).

### Lyar is a Tak1-interacting protein that modulates the proliferation and differentiation of BMMSCs

We previously demonstrated that Tak1 plays a pivotal role in regulating the proliferation and differentiation of BMMSCs [10]. To identify potential molecular partners that link Tak1 activity to these cellular processes, we conducted immunoprecipitation (IP) followed by proteomic analysis using liquid chromatography–mass spectrometry (LC-MS). Given that Tak1 is highly sensitive to cellular stress, we generated a stable embryonic stem (ES) cell line expressing HA-tagged Tak1 to ensure experimental reproducibility and minimize stress-related artifacts associated with transient transfection. From the proteomic dataset, 146 proteins met the criteria of an IP abundance exceeding $1 \times 10^6$ and an enrichment ratio greater than 10-fold (S3 Table). Focusing on molecules with known roles in stem cell proliferation and differentiation, we identified Lyar as a candidate protein previously reported to be involved in cell proliferation (Fig 2A). To elucidate the function of Lyar in BMMSCs, we performed gene silencing using siRNA targeting *Lyar*. The knockdown efficiency was approximately 80% (Fig 2B and 2C), and *Lyar*-silenced BMMSCs exhibited significantly reduced proliferation compared to scrambled RNA controls (Fig 2D). Upon dexamethasone-induced differentiation, *siLyar*-transfected BMMSCs showed upregulated

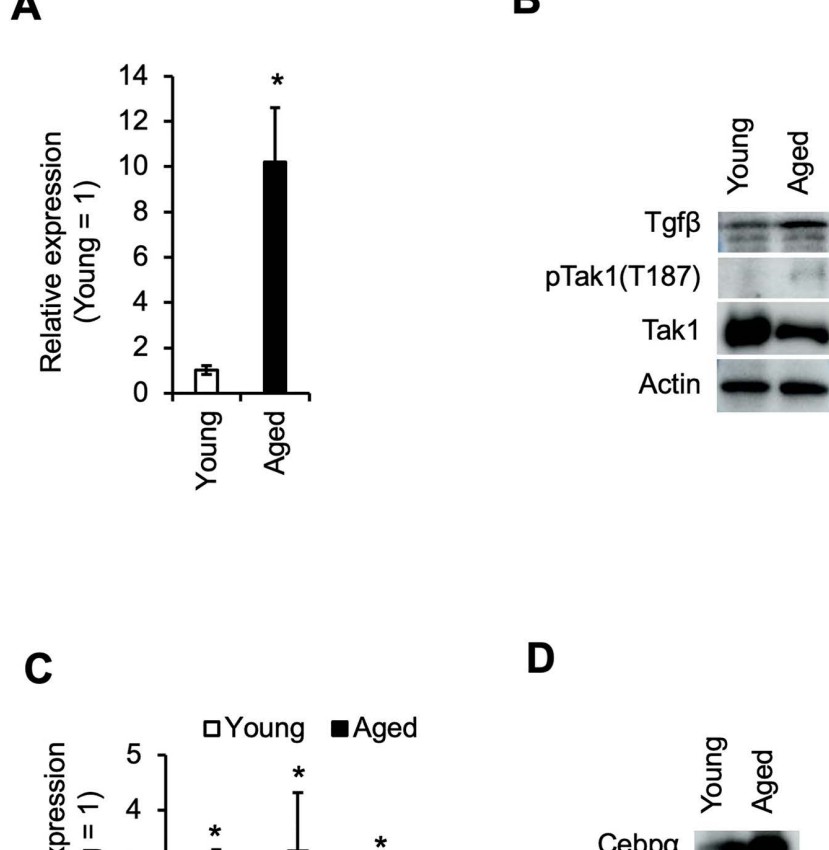

**Fig 1. Expression of TGF-β1 and adipogenic markers is elevated in the bone marrow of aged mice.** A. Quantitative RT-PCR analysis of *Tgfb1* expression in bone marrow from young (3-week-old) and aged (2-year-old) mice. Bars represent the mean±standard deviation of three biological replicates. Asterisks indicate statistically significant differences between groups (*P*<0.05). B. Western blot analysis of bone marrow lysates from young and aged mice. Aged samples show increased expression of TGF-β1 and elevated phosphorylation of Tak1. C. Gene expression levels of adipogenic markers (*Cebpa, Fabp4, Pparg*) in bone marrow from young and aged mice. Bars represent the mean±standard deviation of three biological replicates. Asterisks indicate statistically significant differences (*P*<0.05). D. Western blot analysis showing increased protein levels of adipogenic markers in the bone marrow of aged mice compared to young controls.

expression of both *Runx2*, a master regulator of osteogenesis, and *Pparγ*, a key transcription factor involved in adipogenesis (Fig 2E).    To further investigate the role of Lyar in adipogenic differentiation, we overexpressed Lyar in MSCs (Fig 2F) and induced adipogenesis over a 10-day period. Under these conditions, expression of adipocyte marker genes was suppressed (Fig 2G), and lipid droplet accumulation was markedly decreased (Fig 2H), suggesting that Lyar negatively regulates adipogenic commitment in BMMSCs.

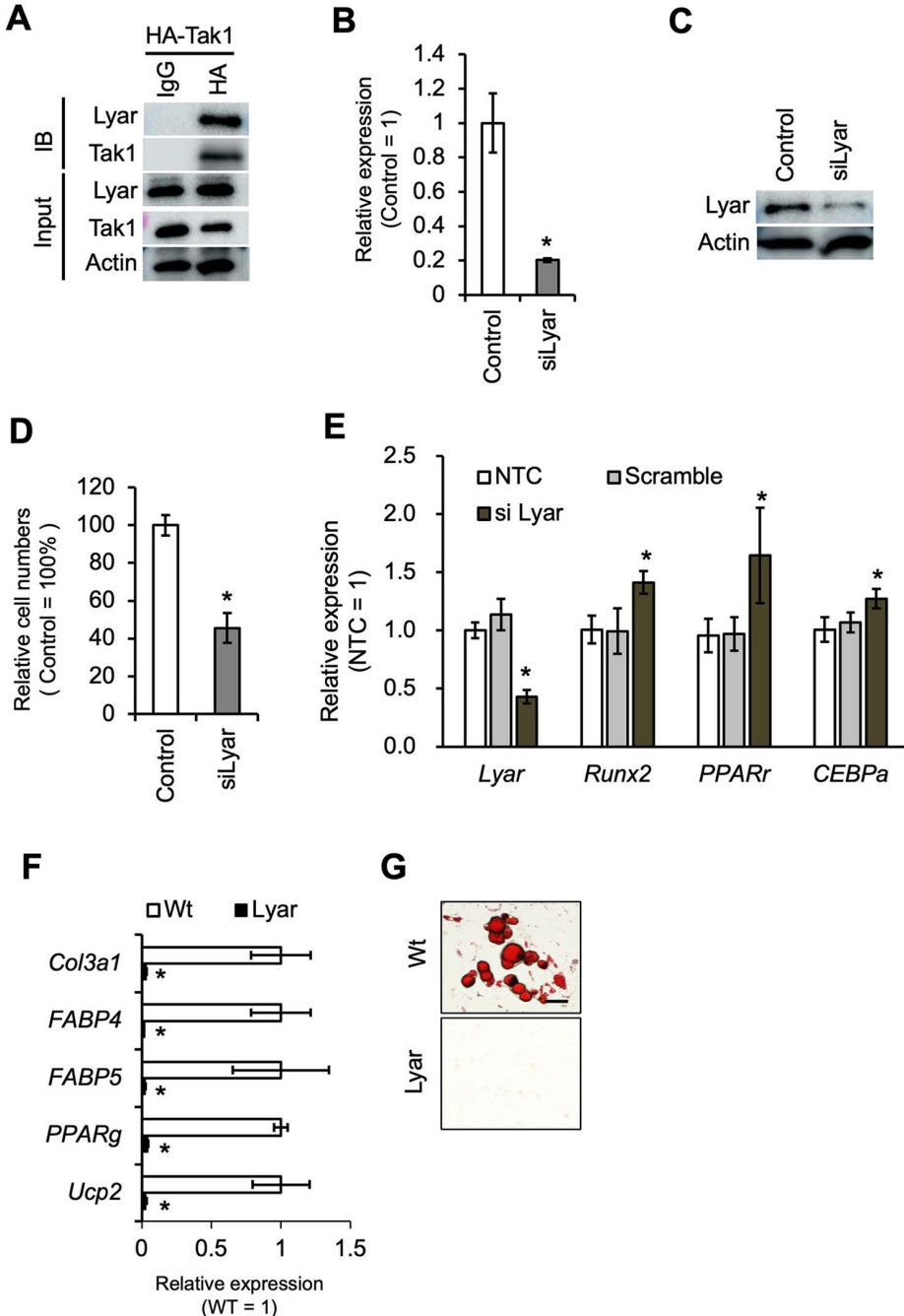

**Fig 2. Lyar regulates cell proliferation and differentiation of BMMSCs.** A. Immunoprecipitation (IP) and immunoblotting (IB) analysis showing the interaction between Tak1 and Lyar. HA-tagged Tak1 was immunoprecipitated using an anti-HA antibody, and co-precipitated proteins were detected using specific antibodies. B. Knockdown efficiency of Lyar in proliferating BMMSCs using siRNA, assessed by quantitative RT-PCR. Lyar expression levels were normalized to GAPDH and quantified using the ΔΔCt method, with the control sample serving as the reference. Bars represent the mean±standard deviation of three biological replicates. Asterisks indicate statistically significant differences between groups ($P < 0.05$). C. Western blot analysis confirming the reduction of Lyar protein levels in BMMSCs transfected with siLyar. D. Knockdown efficiency of Lyar during BMMSC differentiation, as measured by qPCR. Lyar expression levels were normalized to GAPDH and quantified using the ΔΔCt method, with the control sample serving as the reference. Bars represent the mean±standard deviation of three biological replicates. Asterisks indicate statistically significant differences between groups ($P < 0.05$). E. Gene expression analysis of osteogenic marker *Runx2* and adipogenic markers *PPARγ* and *C/EBPα* in BMMSCs treated with siLyar during early differentiation. Expression levels of each gene were normalized to GAPDH and quantified using the ΔΔCt method, with the control sample

serving as the reference. Bars represent the mean±standard deviation of three biological replicates. Asterisks indicate statistically significant differences between groups (P<0.05). F. Gene expression levels of adipogenic markers in BMMSCs overexpressing FLAG-Lyar after 10 days of adipogenic induction. Bars represent the mean±standard deviation of three biological replicates. Expression levels of each gene were normalized to GAPDH and quantified using the ΔΔCt method, with the control sample serving as the reference. Asterisks indicate statistically significant differences between groups (P<0.05). G. Oil Red O staining showing lipid droplet formation as an indicator of adipogenic differentiation. BMMSCs overexpressing Lyar displayed reduced lipid accumulation compared to controls. Scale bar, 20 µm.

## Lyar is involved in the regulation of adipogenic differentiation and interacts with Brd2, a known suppressor of adipogenesis, as one of its underlying mechanisms

Although Tak1 is known to regulate stem cell self-renewal, it has also been reported to promote adipogenesis [14,15], suggesting a dual and context-dependent role. To investigate whether Lyar modulates this functional dichotomy, we overexpressed Tak1 in BMMSCs while simultaneously suppressing Lyar expression using siRNA. Tak1 overexpression alone inhibited adipogenic differentiation; however, co-treatment with siLyar reversed this effect (Fig 3A). These findings suggest that Lyar is a critical modulator in the signaling cascade that governs the balance between MSC self-renewal and adipogenic commitment. To elucidate the molecular mechanism by which Lyar influences adipogenic differentiation, we performed IP using anti-FLAG antibodies in MSCs overexpressing FLAG-tagged Lyar, followed by LC-MS-based proteomic analysis. Among the identified interactors, 1,228 proteins were enriched more than 20-fold relative to the IgG control. Of these, 11 were classified as transcriptional regulators associated with stemness or lineage commitment (Fig 3B). Notably, Brd2 was identified as a putative Lyar-binding partner among the differentiation associated molecules. To validate this interaction, we repeated the IP assay in BMMSCs overexpressing FLAG-Lyar. Co-immunoprecipitation (IP) followed by Western blot analysis in wild-type BMMSCs using an anti-Lyar antibody, performed to confirm that the interaction occurs at endogenous protein levels, supporting a physical association between Lyar and Brd2 in MSCs (Fig 3C and S4 Table). To determine whether Brd2 contributes to Lyar-mediated regulation of adipogenic differentiation, BMMSCs stably overexpressing Lyar were generated using a transposon-based system. These cells were subjected to Brd2 knockdown by siRNA followed by adipogenic induction. At 48 h after the initiation of adipogenic differentiation, the expression levels of the adipogenic master regulators PPARγ and C/EBPα were evaluated. Brd2 suppression resulted in a modest but detectable increase in the expression of these genes (Fig 3D).

## Lyar expression is downregulated in BMMSCs from aged mice and is regulated by Akt signaling

To evaluate the expression of Lyar in BMMSCs from aged mice, we first isolated MSCs using FACS, defining the population as CD45⁻/Ter119⁻/PDGFRα⁺/Sca1⁺. In aged mice, Lyar expression in BMMSCs was approximately 50% at the mRNA level compared to young controls, and protein levels were also markedly decreased (Fig 4A and 4B). To identify upstream regulators of Lyar expression, we conducted a proteomic analysis using BMMSCs derived from both young and aged mice. In senescent BMMSCs derived from aged mice, we observed elevated expression of molecular markers associated with cellular aging and senescence, including H2AX, a key indicator of DNA damage, and Notch2, which has been implicated in age-related alterations in stem cell function and differentiation [16]. Additional upregulated markers included Ctnna1 [17], Pdk1 [18], Stat3 [19], Itgb2 [20], Cdk2 [21], Cdkn2c, Hmgb1, all of which have been linked to senescence-related signaling pathways. In addition to the reduced expression of Lyar, we also observed downregulation of Akt1 and Hmgb2, both of which are associated with stemness (S5 Table). It is well established that FGF2 signaling activates AKT in BMMSCs. Based on this, we hypothesized that the decreased expression of Lyar in aged BMMSCs may be attributed to impaired signaling through the FGF2–PI3K–Akt1 pathway. Supporting this notion, TGFβ treatment had no significant effect on Lyar expression (Fig 4C), whereas FGF2 stimulation increased Lyar expression. In contrast, treatment with a PI3K inhibitor reduced Lyar levels (Fig 4D). siRNA-mediated knockdown of Akt1, achieving approximately

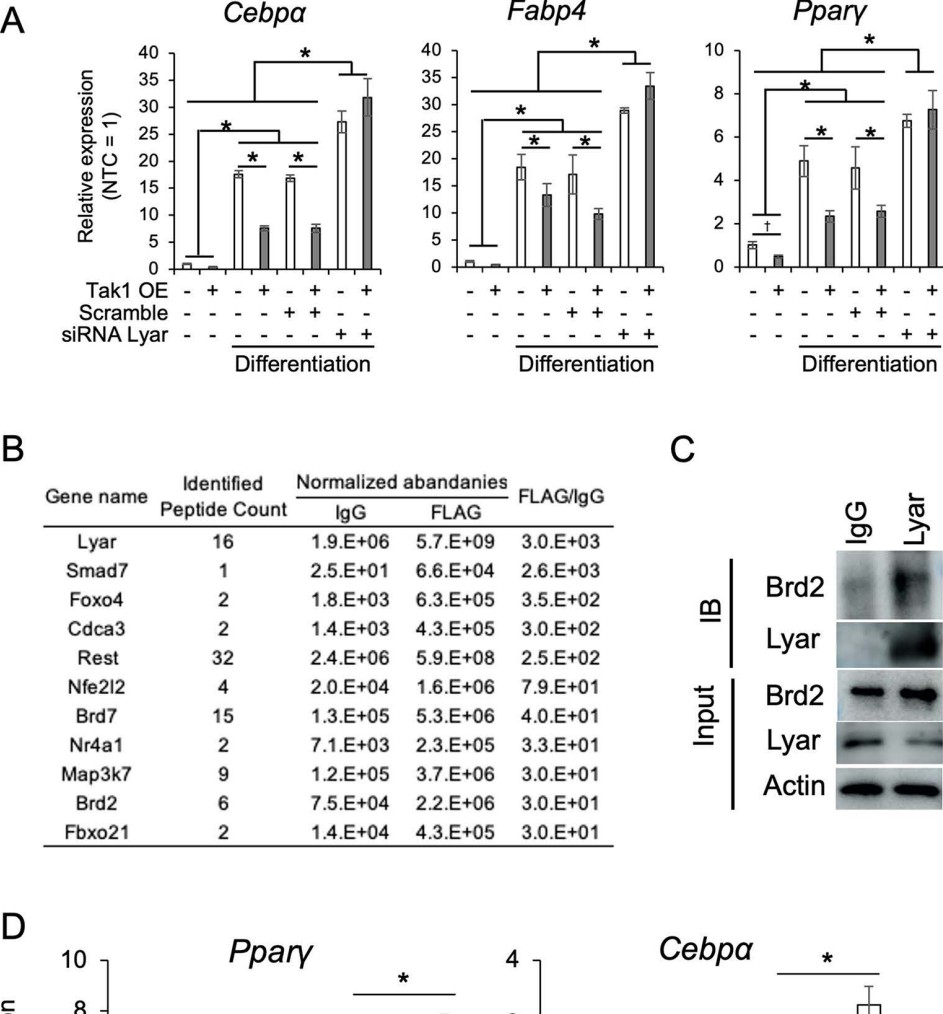

**Fig 3. Lyar interacts with Brd2, a suppressor of adipogenic differentiation.** A. Adipogenic differentiation of BMMSCs under conditions of Tak1 over-expression (Tak1 OE) with or without siLyar. The inhibitory effect of Tak1 OE on adipogenesis was abolished when Lyar was suppressed, suggesting that Lyar is essential for suppression of adipogenesis. Expression levels of each gene were normalized to GAPDH and quantified using the ΔΔCt method, with the control sample serving as the reference. Bars represent the mean±standard deviation of three biological replicates. Asterisks indicate statisti-cally significant differences between groups (*P* < 0.05). B. List of candidate Lyar-interacting proteins identified by immunoprecipitation followed by mass spectrometry (IP-MS) in BMMSCs overexpressing FLAG-tagged Lyar. C. Validation of the interaction between Lyar and Brd2 by immunoprecipitation and Western blotting (IP-WB). Lyar was immunoprecipitated using an anti-Lyar antibody, and the presence of Brd2 in the precipitate was confirmed by immu-noblotting. D. Adipogenic differentiation and the effects of Brd2 suppression in Lyar-overexpressing BMMSCs. Brd2 knockdown partially increased the

expression of the early adipogenic markers Pparγ and Cebpa in Lyar-overexpressing MSCs. Expression levels of each gene were normalized to GAPDH and quantified using the ΔΔCt method, with the control sample serving as the reference. Bars represent the mean ± standard deviation of three biological replicates. Asterisks indicate statistically significant differences between groups ($P < 0.05$).

60% suppression, led to an approximately 20% decrease in Lyar expression (Fig 4E). This decrease was also confirmed at the protein level (Fig 4F).

## Discussion

Aging associated MAT accumulation could be a critical factor for various defects disturbing healthy life in the elderly society [6,22]. Previous studies using senescence-accelerated mouse models (SAMP6) [23] as well as normally aged mice [24], have reported increased adipogenic differentiation with age. Similarly, human CD271+SSEA-4+BMMSCs isolated from elderly individuals exhibit hallmarks of cellular senescence, including diminished proliferative capacity, elevated senescence-associated β-galactosidase activity, impaired osteogenic differentiation, and enhanced adipogenic potential [25]. Numerous reports have demonstrated increased expression of TGFβ in aged tissues [26,27]. In the aging bone marrow microenvironment, senescent cells secrete elevated levels of TGFβ, which disrupts MSC function and impairs hematopoiesis [27,28]. TGFβ1 is also recognized as a major component of the SASP and plays a critical role in mediating the bystander effect through paracrine signaling [29,30].

Although TGFβ has been shown to inhibit adipogenic differentiation [31,32], its regulatory effects are thought to be modulated by interactions with multiple signaling pathways, forming a complex and context-dependent network. We previously identified Tak1 as a critical effector of non-canonical TGFβ signaling that supports BMMSC self-renewal [10]. In the present study, we observed increased Tgfb1 expression and elevated Tak1 phosphorylation in bone marrow tissues from aged mice. In line with previous findings, markers of adipogenic differentiation were also upregulated. While Tak1 is known to promote BMMSC proliferation, it has paradoxically also been reported to facilitate adipogenic differentiation, indicating a dual role in regulating stem cell fate decisions.

It is paradoxical that Tak1, a molecule known to support the self-renewal of stem-like cells such as BMMSCs and satellite cells, has also been reported to promote cellular differentiation. Based on this functional dichotomy, we hypothesized that the direction of Tak1 activity may be regulated through interactions with specific molecular partners. Such regulatory mechanisms may contribute to the regulation of self-renewal and differentiation in stem cell populations.

To explore this possibility, we performed IP–MS analysis to identify molecules that interact with Tak1 and are potentially involved in proliferation or transcriptional regulation. This approach yielded several candidate interactors. Among these, we focused our subsequent analyses on Lyar, a molecule previously reported to play a critical role in stem cell self-renewal [33–36]. Lyar has been reported to show high expression during embryonic development and in ESCs and regulates cell growth and tumorigenesis [33–36]. Consistent with previous reports, Lyar knockdown in BMMSCs led to a significant reduction in proliferation. The impact of Lyar knockdown on cell proliferation and self-renewal capacity was similarly observed in Tak1-overexpressing BMMSCs (S1 Fig). These results suggest that Lyar may be involved in maintaining the stem-like state of BMMSCs.

To further investigate whether Lyar may be involved in regulation of proliferation and differentiation, we transfected mouse BMMSCs with Lyar-targeting siRNA and induced differentiation with dexamethasone for 48 hours and observed increased expression of both Runx2, a key regulator of osteogenesis, and Pparγ, the master transcription factor for adipogenesis. These findings suggest that suppression of Lyar may be associated with a shift toward a more differentiation-commitment state, although alternative interpretations, such as a loss of stem-like properties in BMMSCs, cannot be excluded. Consistent with this, forced expression of Lyar in BMMSCs reduced both osteogenic and adipogenic differentiation efficiency, with a more pronounced inhibitory effect on adipogenesis. Although the extent to which these mechanisms

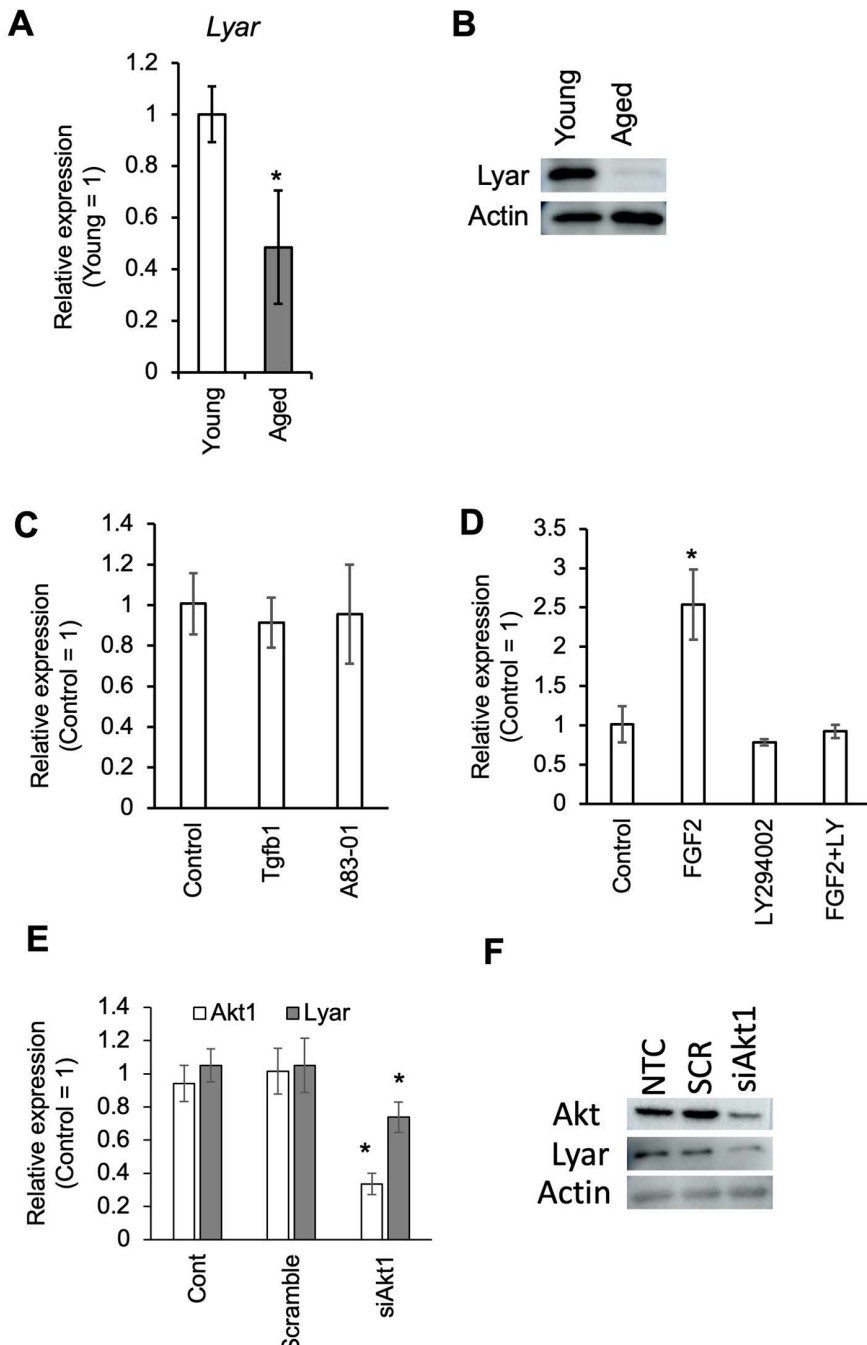

**Fig 4. Downregulation of Lyar expression in aged BMMSCs is induced by dysfunction of the FGF2–PI3K–Akt1 signaling pathway.** A. Comparison of Lyar mRNA expression in BMMSCs isolated from young (3-week-old) and aged (2-year-old) mice. Lyar expression levels were normalized to GAPDH and quantified using the ΔΔCt method, with the control sample serving as the reference. Bars represent the mean ± standard deviation of three biological replicates. Asterisks indicate statistically significant differences between groups ($P < 0.05$). B. Immunoblot analysis showing Lyar protein expression in BMMSCs from young and aged mice. A marked reduction in Lyar levels is observed in aged BMMSCs. C. Effects of TGF-β1 and A83-01 (a TGF-β receptor inhibitor) treatment on Lyar expression in cultured mouse BMMSCs after 48 hours. Neither treatment significantly altered Lyar expression. Lyar expression levels were normalized to GAPDH and quantified using the ΔΔCt method, with the control sample serving as the reference. Bars represent the mean ± standard deviation of three biological replicates. Asterisks indicate statistically significant differences between groups ($P < 0.05$). D. Effects of FGF2 and LY294002 (a PI3K inhibitor) on Lyar expression in cultured mouse BMMSCs. FGF2 treatment increased Lyar expression, whereas

LY294002 reduced its expression, indicating regulation via the PI3K–Akt1 pathway. Lyar expression levels were normalized to GAPDH and quantified using the ΔΔCt method, with the control sample serving as the reference. Bars represent the mean ± standard deviation of three biological replicates. Asterisks indicate statistically significant differences between groups ($P < 0.05$). E. Effect of siRNA-mediated Akt1 knockdown on Lyar expression. Lyar expression levels were normalized to GAPDH and quantified using the ΔΔCt method, with the control sample serving as the reference. Bars represent the mean ± standard deviation of three biological replicates. Asterisks indicate statistically significant differences between groups ($P < 0.05$). F. Effect of Akt1 siRNA (siAkt1) on the protein expression levels of Akt1 and Lyar, as detected by WB analysis.

operate in primary BMMSCs remains to be determined, similar effects were observed in 3T3-L1 cells (S2 Fig). However, additional quantification in 3T3-L1 cells does not fully compensate for the limited quantitative support in primary BMMSCs, and therefore conclusions based on representative images should be interpreted with caution.

Until now, at least three molecular mechanism has been suggested about the function of Lyar as a transcription regulator: LYAR repress human γ-globin by binding to the consensus sequence GGTTAT on 5'-untranslated region of the γ-globin gene with the methyltransferase PRMT5, which triggers the histone H4 Arg3 symmetric dimethylation (H4R3me2s) mark [37]. On the other hand, transcription of IFN-β and its downstream gens were regulated through impeding the DNA binding capacity of IRF3 by Lyar [38]. In the ESCs, Lyar involves regulation of differentiation by recruiting histone-associated protein BRD2 [39] to the promoter of Nanog, which is a master transcriptional factor for pluripotency [40]. These notion means that Lyar can control various genes through interaction with other transcription factors and the combination seem to be cell type/ tissue specific.

We also found that Lyar can interact with several molecules known to counteract senescence-related phenotypes, including Smad7, Foxo4, and Nfe2l2. Among the identified interactors, Brd2 was of particular interest, as it is known to play a critical role in suppressing adipogenic differentiation. Brd2 is a signal transducer and a nuclear-localized serine/threonine kinase [41,42], and can bind to nucleosomal histone lysine residues with its two bromodomains, particularly acetyl-histone H4 and [43]. Brd2 involves cell differentiation processes such as neuronal cells during mouse CNS development [44]. Importantly, some previous papers demonstrated that Brd2 involves suppression of adipocyte differentiation [45,46]. Wang et al. reported that Brd2 represses Pparγ and inhibits adipogenesis in 3T3-L1 pre-adipocytes [47]. Consistent with this, in our experiments, suppression of Brd2 in Lyar-overexpressing cells resulted in increased expression of the early adipogenic master regulators Pparγ and Cebpα. Based on these findings, the data suggest that Brd2 may partly mediate the inhibitory effect on adipogenic differentiation in BMMSCs through its interaction with Lyar, and that disruption of the Lyar–Brd2 interaction may be associated with adipocyte accumulation in aged tissues. However, our observations were limited to a relatively short time window corresponding to the effective duration of siRNA-mediated knockdown (48 h), and the magnitude of the effect was modest. Therefore, rather than completely abrogating the anti-adipogenic effect of Lyar, long-term suppression of Brd2 is more likely to partially attenuate Lyar-mediated regulation. These findings suggest that Lyar-dependent control of adipogenesis likely involves multiple transcriptional regulators, among which Brd2 represents an important component. To further explore the mechanisms underlying Lyar downregulation in aged tissues, we performed a proteomic analysis of in vitro–cultured BMMSCs and found that Akt1 expression was markedly reduced in aged cells. Akt1 is known to play a critical role in maintaining cellular homeostasis [47–49]. In BMMSCs, FGF2-induced PI3K–Akt1 signaling is essential for cell proliferation and stemness maintenance [50,51]. FGF2 treatment increased Lyar expression, while the PI3K inhibitor LY294002 reduced it. In contrast, TGF-β and its receptor inhibitor A83-01 had no significant effect on Lyar levels. Furthermore, siRNA-mediated suppression of Akt1 also reduced Lyar expression. Although the decrease in Lyar expression was modest (approximately 20%), this limited effect may be attributable to the partial nature of Akt1 inhibition in our experiments, where knockdown efficiency reached approximately 70% and residual protein was still detectable. Thus, Akt1 suppression may not have been sufficient to induce a critical reduction in Lyar levels. Nevertheless, a consistent trend indicating a positive association between PI3K–Akt1 signaling and Lyar expression was observed. These findings suggest that age-related attenuation of FGF2–Akt1 signaling may be associated with reduced Lyar expression in aged BMMSCs.

In this study, we investigated the paradoxical observation that adipogenic differentiation is promoted in aged bone marrow, despite the presence of abundant TGFβ signaling, which is generally known to suppress adipogenesis. Focusing on the non-canonical pathway mediator Tak1, we identified the partner protein Lyar as a candidate modulator of this process. Our findings suggest that Lyar may be involved in the suppression of adipogenic differentiation. One possible interpretation is that when both Tak1 and Lyar are present, cells preferentially maintain a proliferative state. In contrast, in the absence of Lyar, Tak1 may shift toward supporting adipogenic differentiation. In this framework, the functional outcome of Tak1 signaling may be influenced by its interaction with Lyar in a context-dependent manner, thereby contributing to the age-related shift in MSC fate. Notably, Tak1 loss in adipocytes has been reported to reduce adipocyte numbers and promote browning of white adipose tissue [14]. Gallot et al. demonstrated that Tak1 deletion improves metabolic outcomes by reducing diet-induced obesity and enhancing energy expenditure [52]. In contrast, our previous work observed a slight increase in adipogenesis upon Tak1 inhibition in BMMSCs [10].

Finally, our study has several limitations.

First, Tak1 exerts diverse biological functions and interacts with a wide range of molecular partners; therefore, the mechanism described here involving Lyar likely represents only one of several regulatory pathways governing Tak1-dependent outcomes. Moreover, because Lyar is closely associated with cell proliferation, its sustained or unregulated activation may increase the risk of oncogenic transformation, as suggested by previous studies. Although our data indicate a functional relationship between Tak1 and Lyar, they provide only partial evidence that Lyar regulates Tak1 activity, and further mechanistic studies will be required to clarify this relationship. In addition, because Tak1 can be activated by signals other than TGFβ, therapeutic strategies targeting the TGFβ–Tak1–Lyar axis should be interpreted with careful consideration of the broader regulatory context of Tak1 signaling.

Second, this study used GAPDH as the sole reference gene for qPCR normalization. In aging-related mesenchymal systems, the stability of GAPDH expression should not be assumed without validation. Although the age-associated decline in Lyar expression is supported by protein and proteomic data, several mechanistic interpretations rely on qPCR-based measurements and should therefore be interpreted with caution.

Third, it remains unclear whether reduction of Lyar constitutes a critical driver of age-related marrow adiposity. In the present study, sustained suppression of Lyar in BMMSCs was technically difficult, limiting our ability to evaluate late-stage adipogenic maturation. Future studies using conditional knockout or inducible transgenic models will be important to more objectively assess the contribution of Lyar to tissue integrity and age-associated adipogenic changes.

Fourth, validation in human systems is required. Although Lyar is conserved in humans, it remains unclear whether the same regulatory mechanisms operate and whether Tak1–Lyar interactions function similarly in human MSCs. Further investigation will be required to address these issues.

In conclusion, we demonstrated here that the nucleolar protein Lyar may be involved in the regulation of adipocyte differentiation, potentially in association with Brd2. However, the biological role of Lyar in tissue development and homeostasis remains poorly understood, as Lyar-deficient mice are viable and fertile with no overt phenotype [33,53]. Further investigations using more refined approaches, possibly informed by gerontological perspectives, may offer new insights into the mechanisms of age-related tissue dysfunction.

## Supporting information

**S1 Fig. siRNA-mediated suppression of Lyar in Tak1-overexpressing MSCs and its impact on MSC self-renewal.** (JPEG)

**S2 Fig. Lyar overexpression suppresses adipogenic differentiation.** (JPEG)

**S3 Fig. Uncropped Western blot images corresponding to the main figures.** (PDF)

**S1 Table. Primer sequences used in qRT-PCR analysis.**
(PDF)

**S2 Table. Antibodies used in FACS and WB analyses in the present study.**
(PDF)

**S3 Table. List of proteins interacting with Tak1 identified by mass spectrometry following immunoprecipitation of HA-Tak1.**
(PDF)

**S4 Table. List of proteins interacting with Lyar identified by mass spectrometry following immunoprecipitation of FLAG-Lyar.**
(PDF)

**S5 Table. List of proteins differentially expressed in BMMSCs from young and aged mice identified by shotgun proteomics.**
(PDF)

## Author contributions

**Conceptualization:** Yu Shinyashiki, Yuta Onodera, Natsumi Iwawaki, Koji Goto, Takeshi Teramura.

**Data curation:** Yu Shinyashiki, Yuta Onodera, Yusuke Kawashima, Natsumi Iwawaki, Toshiyuki Takehara.

**Formal analysis:** Yu Shinyashiki, Yuta Onodera, Natsumi Iwawaki.

**Funding acquisition:** Takeshi Teramura.

**Investigation:** Yu Shinyashiki, Yuta Onodera, Yusuke Kawashima, Natsumi Iwawaki, Takeshi Teramura.

**Methodology:** Yuta Onodera, Yusuke Kawashima, Natsumi Iwawaki.

**Project administration:** Takeshi Teramura.

**Resources:** Yu Shinyashiki, Yuta Onodera.

**Software:** Toshiyuki Takehara.

**Supervision:** Yusuke Kawashima, Koji Goto.

**Validation:** Yuta Onodera, Toshiyuki Takehara, Koji Goto.

**Writing – original draft:** Yu Shinyashiki, Yuta Onodera, Natsumi Iwawaki, Takeshi Teramura.

**Writing – review & editing:** Toshiyuki Takehara, Koji Goto, Takeshi Teramura.

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
