## [Decision Letter · Decision Letter 0]

20 Nov 2025

PONE-D-25-42314Decreased expression of Ly-1 antibody reactive clone (Lyar) triggers enhanced adipogenesis of bone marrow mesenchymal stromal cells in aged bone marrowPLOS ONE

Dear Dr. Teramura,

Thank you for submitting your manuscript to PLOS ONE. After careful consideration, we feel that it has merit but does not fully meet PLOS ONE’s publication criteria as it currently stands. Therefore, we invite you to submit a revised version of the manuscript that addresses the points raised during the review process.

We look forward to receiving your revised manuscript.

Kind regards,

Sadiq Umar

Academic Editor

PLOS ONE

Journal Requirements:

**Additional Editor Comments:**

Please include data demonstrating FGF2–PI3K–Akt1 signaling activity in Figure 4. Without showing this signaling cascade, it is premature to draw conclusions about its functional involvement.

Reviewers' comments:

Reviewer's Responses to Questions

**Comments to the Author**

1. Is the manuscript technically sound, and do the data support the conclusions?

Reviewer #1: Yes

Reviewer #2: Yes

2. Has the statistical analysis been performed appropriately and rigorously? 

Reviewer #1: Yes

Reviewer #2: No

3. Have the authors made all data underlying the findings in their manuscript fully available?

Reviewer #1: Yes

Reviewer #2: Yes

4. Is the manuscript presented in an intelligible fashion and written in standard English?

Reviewer #1: Yes

Reviewer #2: Yes

5. Review Comments to the Author

Reviewer #1: This study identifies the nucleolar protein Lyar as a novel interacting partner of TGF-β activated kinase 1 (Tak1) and demonstrates its critical role in suppressing adipogenic differentiation of bone marrow mesenchymal stromal cells (BMMSCs) during aging. The authors present a compelling model wherein age-related downregulation of Lyar, driven by diminished FGF2–PI3K–Akt1 signaling, disrupts the balance of the TGF-β–Tak1 axis and promotes marrow adipose tissue (MAT) accumulation. The work provides valuable insights into the molecular mechanisms of aging in the bone marrow niche. However, several issues require attention to strengthen the study's conclusions and translational relevance.

1.Incomplete Mechanistic Elucidation of the Lyar–Brd2 Axis. While the interaction between Lyar and Brd2 is demonstrated, the precise molecular mechanism by which this complex suppresses adipogenesis is not defined.

2. Ambiguous Functional Interplay Between Tak1 and Lyar. The model posits that Lyar determines the functional output of Tak1 signaling, but direct evidence is limited. The biochemical consequence of the Lyar–Tak1 interaction on Tak1's kinase activity or its downstream non-canonical pathways (e.g., p38/JNK activation) is not investigated, leaving the proposed "switch" mechanism speculative.

3.Lack of Rescue Experiments. The most functionally compelling evidence—demonstrating that forced expression of Lyar in aged BMMSCs can rescue the hyper-adipogenic phenotype—is absent. Such an experiment is critical to substantiate the claim that Lyar downregulation is a causative factor in age-enhanced adipogenesis.

4.Inadequate Discussion of the Broader Field and Study Limitations. The discussion does not sufficiently contextualize the findings within the existing literature on TGF-β signaling in stem cell aging. Furthermore, the limitations of the study, particularly the sole reliance on murine models and the preliminary nature of the mechanistic insights, should be explicitly acknowledged.

Reviewer #2: The study offers valuable insights into the potential roles of Lyar and Tak1 in age-associated adipogenic changes in bone marrow. However, portions of the Discussion overinterpret the current data. Refining the claims to better align with the presented evidence, and clearly distinguishing between demonstrated findings and hypotheses, will enhance the manuscript’s clarity and impact.

M&M

Overall, there is no description of the protocols, equipment, or detectors used. Furthermore, when protocols are not described, the references provided are insufficient to enable replication.

1. Quantitative RT-PCR (qRT-PCR) and Western blot (WB) analysis

The quality of RNA, reverse transcription and qPCR conditions, as well as housekeeping genes such as GAPDH used for quantification, are often unsuitable for studies involving aging.

2. Treatment of primary MSCs with siRNA

The current description lacks information on cell number, final siRNA concentration, the ratio to RNAiMAX, and the negative control siRNA, making it difficult for third parties to reproduce the conditions exactly. These parameters must be added.

3.Immunoprecipitation (IP)

I understand the procedure flow, but the composition of the lysis and washing buffers, the total protein amount and antibody amount per IP, and the reaction volume are not specified. This makes it difficult for other facilities to replicate the same conditions. I believe these parameters should be added.

4. On-bead digestion of IP samples for proteome analysis

This procedure is generally described in a reproducible manner, but having information on the reaction solution volume, StageTip washing/elution solvent composition, and protein content in the IP sample (relative to enzyme amount) would make it easier for others to reproduce it more rigorously

5. Mass spectrometry of IP samples using anti-HA (HA-Lyar) antibody

Regarding the LC–MS analysis of anti-HA IP samples, critical information necessary for ensuring reproducibility is significantly lacking. Specifically, the mobile phase composition, gradient conditions, flow rate, column temperature, injection volume, MS1/MS2 scan parameters, NCE, isolation width, and other details are all missing. Furthermore, information such as the precursor/fragment tolerance, fixed/variable modifications, and number of miscleavages used in the SEQUEST analysis is also required. Without this information, third-party replication of the experiments is difficult; therefore, these details should be added.

6. The manuscript does not clearly describe the number of biological replicates used in each experiment, nor does it consistently report the statistical methods applied. To ensure transparency and reproducibility, I recommend adding a dedicated “Statistical Analysis” section at the end of the Methods. This section should specify the number of biological replicates for each assay, the statistical tests used, any adjustments for multiple comparisons, and the software employed for data analysis.

Result

Lyar is a Tak1-interacting protein that modulates the proliferation and differentiation of BMMSCs

7. We previously demonstrated that Tak1 plays a pivotal role in regulating the proliferation and differentiation of BMMSCs.

Please cite the literature.

8. we generated a stable embryonic stem (ES) cell line expressing HA-tagged Tak1 to ensure experimental reproducibility and minimize stress-related artifacts associated with transient transfection.

A description or citation is required regarding the creation of stable embryonic stem (ES) cell lines expressing Tak1 tagged with HA in M&M.

9. Focusing on molecules with known roles in stem cell proliferation and differentiation, we identified Lyar as a candidate protein previously reported to be involved in cell proliferation (Figure 2A)

The authors state that they “focused on molecules with known roles in stem cell proliferation and differentiation, identifying Lyar as a candidate from among them.” However, the 146 proteins listed in the Supplementary file appear to include other nuclear factors similarly reported to be involved in stem cell properties or differentiation control. Providing more specific reasons for prioritizing Lyar for analysis—such as the strength of enrichment, prior reports in MSCs/ESCs, or specific literature context—would clarify the rationale for this candidate selection.

Lyar expression is downregulated in BMMSCs from aged mice and is regulated by Akt signaling

10. hypothesized that the decreased expression of Lyar in aged BMMSCs may be attributed to impaired signaling through the FGF2–PI3K–Akt1 pathway

Although it has been suggested that FGF2–PI3K–Akt1 may positively regulate Lyar expression, the current data remain at the correlation level, and additional verification is needed to conclude a direct mechanistic link.

For example, confirming the response of Lyar expression to forced expression of Akt1 or siAkt1 would likely strengthen the argument.

Discussion

This study provides an interesting exploration of the molecular basis underlying the adipogenic shift observed in aged bone marrow, focusing particularly on the roles of Tak1, Lyar, and Brd2. The identification of Lyar as a potential regulator in this context is noteworthy. However, several aspects of the Discussion extend beyond the evidence presented in the Results, and some conclusions appear stronger than what the current data can support. Clarification and refinement of several key points would substantially strengthen the overall interpretation of the study. Major concerns are summarized below.

11. The dual and context-dependent roles of Tak1 require clearer explanation

The Discussion highlights that Tak1 supports stem cell self-renewal but has also been reported to promote adipogenesis. In contrast, the current study shows that Tak1 overexpression suppresses adipogenic differentiation (Fig. 3A). The relationship between these findings and previously reported functions is not fully reconciled. A more explicit interpretation of how the current results fit into the broader context of Tak1 biology would improve clarity.

12. Interpretation of Lyar knockdown effects appears overstated

In Fig. 2E, both Runx2 and Pparγ are upregulated 48 hours after Lyar knockdown. The Discussion interprets this as evidence that Lyar suppression promotes lineage commitment. However, simultaneous upregulation of early osteogenic and adipogenic markers at such an early time point may instead reflect loss of stemness or a general differentiation-prone state, rather than commitment toward a specific lineage. Although the manuscript includes functional adipogenesis assays for Lyar overexpression (Fig. 2H), no functional readouts are provided for the knockdown condition. Without later-stage adipogenic markers or lipid accumulation data under knockdown, commitment cannot be conclusively inferred.

13. The claim that Lyar regulates “the balance between self-renewal and adipogenic commitment” is not fully supported

Fig. 3A demonstrates that Lyar knockdown abolishes the anti-adipogenic effect of Tak1 overexpression. While this suggests involvement in adipogenic regulation, the study does not present data assessing self-renewal or stemness under these conditions.

Therefore, concluding that Lyar governs the “balance” between self-renewal and adipogenic commitment overreaches the available evidence.

14. Functional significance of the Lyar–Brd2 interaction is not demonstrated

The proteomics analysis and co-immunoprecipitation (Fig. 3C) convincingly show a physical interaction between Lyar and Brd2. However, no functional experiments (e.g., Brd2 knockdown, pharmacologic inhibition, or interaction-disruption assays) are provided to demonstrate that this interaction mediates the observed anti-adipogenic effect. Thus, statements implying that Lyar suppresses adipogenesis through Brd2 should be toned down or framed as a hypothesis rather than a demonstrated mechanism.

15. Identification of upstream regulators of Lyar expression remains inconclusive

The proteomic analysis reveals increased senescence-associated markers and decreased Akt1/Hmgb2 levels in aged BMMSCs. However, these results describe the cellular state of aged BMMSCs, rather than identifying direct upstream regulators of Lyar.

The proposed involvement of the FGF2–PI3K–Akt1 pathway is intriguing, establishing causality requires additional experiments (e.g., Akt1 knockdown, constitutively active Akt rescue).

16. The mention of “aging-associated hypermethylation” lacks supporting evidence

The Discussion suggests that reduced Lyar expression in aged tissues may be caused by aging-associated promoter hypermethylation. However, no methylation analysis was performed in this study.

This statement should be rephrased as a potential hypothesis rather than an inferred mechanism.

17. Several conclusions rely on wording that is stronger than the supporting data

Terms such as “regulates,” “governs,” “represses,” and “critical modulator” appear in contexts where the data support only correlation or partial mechanistic involvement. Adjusting these to more cautious terms (e.g., “suggests,” “may contribute to,” or “is consistent with”) would improve scientific rigor.

Figures and Tables

18 Missing methodological details in figure legends

Several figure legends do not provide essential experimental information. For example, Figures 2E and 2G do not specify the normalization gene used for qPCR, and the conditions for adipogenic induction in Figure 3A are not fully described. Including these details directly in each legend would improve clarity and reproducibility.

19. Lack of quantitative data for image-based results

Some panels present only representative images without quantitative analysis, particularly the Oil Red O staining in Figures 2H and 3A and the Western blot in Figure 4B. Providing quantitative assessments—such as ORO absorbance, lipid droplet area, or densitometric analysis—would strengthen the conclusions derived from these figures. Additionally, the Oil Red O staining method should be added to the Materials and Methods section.

20. Absence of scale bars in microscopy images

Several microscopy images appear to lack scale bars, making it difficult to interpret morphological differences across conditions. Adding scale bars to all image panels would enhance interpretability.

21. Insufficient annotation and clarity in Supplementary file 2

The proteomic dataset in Supplementary file 2 does not clearly indicate the thresholds used for differential protein expression, and key proteins highlighted in the main text (e.g., Akt1, Hmgb2, H2AX, Notch2) are not visually emphasized. Improving annotation—such as marking relevant proteins or providing summary plots—would help readers better understand how the proteomic results support the conclusions.

6. PLOS authors have the option to publish the peer review history of their article (what does this mean?). If published, this will include your full peer review and any attached files.

Reviewer #1: No

Reviewer #2: No

---

## [Author Response · Author response to Decision Letter 1]

11 Feb 2026

We appreciate the time and effort devoted to the review of our manuscript despite your busy schedule. As detailed below, we have carefully addressed each of your comments to the fullest extent possible. We would be grateful for the opportunity to have our revised manuscript evaluated again.

Q1. Additional Editor Comments:

Please include data demonstrating FGF2–PI3K–Akt1 signaling activity in Figure 4. Without showing this signaling cascade, it is premature to draw conclusions about its functional involvement.

A1. Your important comment regarding the direct involvement of AKT in regulating Lyar is greatly appreciated. A similar point was also raised by Reviewer 2. In accordance with these comments, we performed AKT1 knockdown and confirmed that Lyar expression decreases as a result. These new experimental data have been added to the revised manuscript as Figures 4E and 4F.

Reviewer 1 comments:

Q1. Incomplete Mechanistic Elucidation of the Lyar–Brd2 Axis. While the interaction between Lyar and Brd2 is demonstrated, the precise molecular mechanism by which this complex suppresses adipogenesis is not defined.

A1. We appreciate this important comment. To address your question, we generated Lyar-overexpressing MSCs and examined the effects of Brd2 knockdown on differentiation. In this experiment, Lyar-overexpressing MSCs were cultured in adipogenic differentiation medium for 48 hours and then analyzed by RT-PCR.

As a result, Brd2 suppression increased the expression of early adipogenic markers such as PPARγ and C/EBPα. Although the inhibitory effect of Lyar on adipogenic differentiation is strong and Brd2 suppression alone was not sufficient to induce full adipogenic differentiation, these findings indicate that Lyar-mediated suppression of adipogenesis can be at least partially rescued by reducing Brd2 expression.

Q2. Ambiguous Functional Interplay Between Tak1 and Lyar. The model posits that Lyar determines the functional output of Tak1 signaling, but direct evidence is limited. The biochemical consequence of the Lyar–Tak1 interaction on Tak1's kinase activity or its downstream non-canonical pathways (e.g., p38/JNK activation) is not investigated, leaving the proposed "switch" mechanism speculative.

A2. We appreciate this important comment. As you pointed out, our study does not clarify how the TAK1–Lyar interaction influences TAK1 activity or other downstream pathways that may be involved. However, Lyar has been reported to participate in cell proliferation and the maintenance of stemness, and in our experiments we demonstrated that TAK1 and Lyar interact, and that suppression of Lyar reduces cellular self-renewal even under conditions of TAK1 overexpression.

That said, as you noted, proposing that Lyar functions as a switch regulating TAK1 activity would be speculative. Therefore, in the revised manuscript we have explicitly added this point as a limitation and clarified that the precise relationship between TAK1 and Lyar remains to be elucidated.

Q3. Lack of Rescue Experiments. The most functionally compelling evidence—demonstrating that forced expression of Lyar in aged BMMSCs can rescue the hyper-adipogenic phenotype—is absent. Such an experiment is critical to substantiate the claim that Lyar downregulation is a causative factor in age-enhanced adipogenesis.

A3. We have already demonstrated that forced expression of Lyar suppresses adipogenic differentiation. Unfortunately, we were unable to perform gene introduction into aged cells followed by drug selection in mouse MSCs. As is well recognized, mouse BM-MSCs are highly vulnerable to stress, and MSCs derived from aged mice in particular have limited proliferative capacity; therefore, our experiments could only be conducted using MSCs from young to adult mice.

As you appropriately noted, it would be important to examine whether senescent phenotypes can be reversed and to more directly demonstrate involvement in age-related marrow adiposity. Ideally, this would require the use of conditional transgenic models. However, generation of transgenic mice was beyond the scope of the present study. We have therefore added this point as a limitation in the revised manuscript.

Q4. Inadequate Discussion of the Broader Field and Study Limitations. The discussion does not sufficiently contextualize the findings within the existing literature on TGF-β signaling in stem cell aging. Furthermore, the limitations of the study, particularly the sole reliance on murine models and the preliminary nature of the mechanistic insights, should be explicitly acknowledged.

A4. We appreciate this comment. The present study focused on TGFβ–TAK1 signaling and Lyar, which is thought to act downstream and to play an important role in the switch between self-renewal and differentiation. As you pointed out, the original wording was indeed unclear; therefore, we have revised the manuscript to improve clarity and added the necessary explanations. In addition, we have carefully incorporated and clearly stated the limitations raised by two reviewers.

Reviewer 2 comments:

Q1. Overall, there is no description of the protocols, equipment, or detectors used. Furthermore, when protocols are not described, the references provided are insufficient to enable replication　(1-5)

A1. We appreciate your careful review. We have revised the manuscript in accordance with your comments.

Q2. The manuscript does not clearly describe the number of biological replicates used in each experiment, nor does it consistently report the statistical methods applied. To ensure transparency and reproducibility, I recommend adding a dedicated “Statistical Analysis” section at the end of the Methods. This section should specify the number of biological replicates for each assay, the statistical tests used, any adjustments for multiple comparisons, and the software employed for data analysis.

A2. In accordance with your suggestion, we have described the statistical analyses in an independent section. We have also specified the software used for the analyses in that section.

Q3. Lyar is a Tak1-interacting protein that modulates the proliferation and differentiation of BMMSCs 7. We previously demonstrated that Tak1 plays a pivotal role in regulating the proliferation and differentiation of BMMSCs.

Please cite the literature.

A3. We sincerely apologize for the oversight. The necessary text has now been appropriately cited.

Q4. we generated a stable embryonic stem (ES) cell line expressing HA-tagged Tak1 to ensure experimental reproducibility and minimize stress-related artifacts associated with transient transfection.

A description or citation is required regarding the creation of stable embryonic stem (ES) cell lines expressing Tak1 tagged with HA in M&M.

A4. We sincerely apologize for the omission of important information in the previous version of the manuscript. In the revised manuscript, we have included a description of the method used to generate HA-tagged TAK1-overexpressing ES cells.

Q5. Focusing on molecules with known roles in stem cell proliferation and differentiation, we identified Lyar as a candidate protein previously reported to be involved in cell proliferation (Figure 2A)

The authors state that they “focused on molecules with known roles in stem cell proliferation and differentiation, identifying Lyar as a candidate from among them.” However, the 146 proteins listed in the Supplementary file appear to include other nuclear factors similarly reported to be involved in stem cell properties or differentiation control. Providing more specific reasons for prioritizing Lyar for analysis—such as the strength of enrichment, prior reports in MSCs/ESCs, or specific literature context—would clarify the rationale for this candidate selection.

Lyar expression is downregulated in BMMSCs from aged mice and is regulated by Akt signaling hypothesized that the decreased expression of Lyar in aged BMMSCs may be attributed to impaired signaling through the FGF2–PI3K–Akt1 pathway

Although it has been suggested that FGF2–PI3K–Akt1 may positively regulate Lyar expression, the current data remain at the correlation level, and additional verification is needed to conclude a direct mechanistic link.

For example, confirming the response of Lyar expression to forced expression of Akt1 or siAkt1 would likely strengthen the argument.

A5. We greatly appreciate this important comment. As you pointed out, the list of proteins shown in the Supplemental file includes other molecules that could potentially be involved in self-renewal and differentiation. Our study was originally initiated to clarify how TAK1 regulates MSC self-renewal. From this perspective, we narrowed our focus to molecules that function in the nucleus as transcription factors or related accessory proteins and that could be directly linked to cellular self-renewal. Among these candidates, Lyar was selected because it is stably expressed at the protein level in MSCs and was reproducibly detected in TAK1 immunoprecipitation experiments.

As a detailed description of this selection process would be overly lengthy for the manuscript, we instead added a concise explanation stating that, among these molecules, we focused on Lyar because it has already been reported to be deeply involved in stem cell self-renewal.

Q10. The dual and context-dependent roles of Tak1 require clearer explanation

The Discussion highlights that Tak1 supports stem cell self-renewal but has also been reported to promote adipogenesis. In contrast, the current study shows that Tak1 overexpression suppresses adipogenic differentiation (Fig. 3A). The relationship between these findings and previously reported functions is not fully reconciled. A more explicit interpretation of how the current results fit into the broader context of Tak1 biology would improve clarity.

A10. We greatly appreciate this important comment. What we intended to emphasize is that whether TAK1 activation leads cells toward proliferation or adipogenic differentiation is highly dependent on the cellular context, and that the presence of Lyar plays a key role in determining this outcome. We believe that Figure 3 illustrates this point.

The present findings underscore the functional importance of TAK1 biology. Beyond its well-established roles in cell proliferation and inflammation, our results suggest that, through interactions with downstream factors, TAK1 may also influence aging-related changes and differentiation.

Q11. Interpretation of Lyar knockdown effects appears overstated

In Fig. 2E, both Runx2 and Pparγ are upregulated 48 hours after Lyar knockdown. The Discussion interprets this as evidence that Lyar suppression promotes lineage commitment. However, simultaneous upregulation of early osteogenic and adipogenic markers at such an early time point may instead reflect loss of stemness or a general differentiation-prone state, rather than commitment toward a specific lineage. Although the manuscript includes functional adipogenesis assays for Lyar overexpression (Fig. 2H), no functional readouts are provided for the knockdown condition. Without later-stage adipogenic markers or lipid accumulation data under knockdown, commitment cannot be conclusively inferred.

A11. We appreciate this important comment. In our experimental system, stable suppression of Lyar using shRNA or gene editing markedly inhibited BM-MSC proliferation, which prevented us from establishing Lyar-deficient cells. Consequently, it was not feasible to maintain Lyar knockdown cells in adipogenic differentiation culture for two weeks, and we were therefore unable to obtain data on late-stage adipogenesis or lipid accumulation as you suggested. We have added this point as a limitation in the revised manuscript.

Q12. The claim that Lyar regulates “the balance between self-renewal and adipogenic commitment” is not fully supported

Fig. 3A demonstrates that Lyar knockdown abolishes the anti-adipogenic effect of Tak1 overexpression. While this suggests involvement in adipogenic regulation, the study does not present data assessing self-renewal or stemness under these conditions.

Therefore, concluding that Lyar governs the “balance” between self-renewal and adipogenic commitment overreaches the available evidence.

A12. We appreciate this important comment. In response to your suggestion, we isolated BM-MSCs from adult mice, established cells with stable TAK1 overexpression, and performed Lyar knockdown. The results were consistent with our previous findings and confirmed that suppression of Lyar reduces BM-MSC self-renewal and proliferation. These data are presented as a Supplemental Figure.

Q13. Functional significance of the Lyar–Brd2 interaction is not demonstrated

The proteomics analysis and co-immunoprecipitation (Fig. 3C) convincingly show a physical interaction between Lyar and Brd2. However, no functional experiments (e.g., Brd2 knockdown, pharmacologic inhibition, or interaction-disruption assays) are provided to demonstrate that this interaction mediates the observed anti-adipogenic effect. Thus, statements implying that Lyar suppresses adipogenesis through Brd2 should be toned down or framed as a hypothesis rather than a demonstrated mechanism.

A13. We appreciate this important comment. A similar concern was also raised by Reviewer 1. To address your question, we generated Lyar-overexpressing MSCs and examined the impact of Brd2 knockdown on differentiation. In this experiment, Lyar-overexpressing MSCs were cultured in adipogenic differentiation medium for 48 hours and subsequently analyzed by RT-PCR.

As a result, suppression of Brd2 increased the expression of early adipogenic markers such as PPARγ and C/EBPα. Although the inhibitory effect of Lyar on adipogenic differentiation is strong and Brd2 suppression alone was not sufficient to induce full adipogenic differentiation, these findings indicate that Lyar-mediated suppression of adipogenesis can be at least partially rescued by reducing Brd2 expression.

Q14. Identification of upstream regulators of Lyar expression remains inconclusive

The proteomic analysis reveals increased senescence-associated markers and decreased Akt1/Hmgb2 levels in aged BMMSCs. However, these results describe the cellular state of aged BMMSCs, rather than identifying direct upstream regulators of Lyar.

The proposed involvement of the FGF2–PI3K–Akt1 pathway is intriguing, establishing causality requires additional experiments (e.g., Akt1 knockdown, constitutively active Akt rescue).

A14. We appreciate this important comment. As you pointed out, our study does not clarify how the TAK1–Lyar interaction affects TAK1 activity or other downstream pathways that may be involved. However, Lyar has already been reported to participate in cell proliferation and the maintenance of stemness, and in our experiments we demonstrated that TAK1 and Lyar interact, and that suppression of Lyar reduces cellular self-renewal even under conditions of TAK1 overexpression.

At the same time, as you noted, proposing that Lyar functions as a switch for TAK1 activity would be speculative. Therefore, in the revised manuscript we have explicitly acknowledged this as a limitation and clarified that the precise relationship between TAK1 and Lyar remains to be elucidated.

Q15. The mention of “aging-associated hypermethylation” lacks supporting evidence

The Discussion suggests that reduced Lyar expression in aged tissues may be caused by aging-associated promoter hypermethylation. However, no methylation analysis was performed in this study.

This statement should be rephrased as a potential hypothesis rather than an inferred mechanism.

A15. We sincerely apologize for the oversight. In accordance with your comment, we have removed the statements regarding age-related methylation.

Q16. Several conclusions rely on wording that is stronger than the supporting data

Terms such as “regulates,” “governs,” “represses,” and “critical modulator” appear in contexts where the data support only correlation or partial mechanistic involvement. Adjusting these to more cautious terms (e.g., “suggests,” “may contribute to,” or “is consistent with”) would improve scientific rigor.

A16. We appreciate this important comment. In ac

---

## [Decision Letter · Decision Letter 1]

21 Apr 2026

PONE-D-25-42314R1Decreased expression of Ly-1 antibody reactive clone (Lyar) triggers enhanced adipogenesis of bone marrow mesenchymal stromal cells in aged bone marrowPLOS One

Dear Dr. Teramura,

Thank you for submitting your manuscript to PLOS ONE. After careful consideration, we feel that it has merit but does not fully meet PLOS ONE’s publication criteria as it currently stands. Therefore, we invite you to submit a revised version of the manuscript that addresses the points raised during the review process.

We look forward to receiving your revised manuscript.

Kind regards,

Gea Oliveri Conti, Ph.D. MBs

Academic Editor

PLOS One

Journal Requirements:

Additional Editor Comments:

We appreciated the improvement of the manuscript, but it still needs some minor revisions. I suggest to authors carefully consider the reviewer's suggestions and comments.

Reviewers' comments:

Reviewer's Responses to Questions

**Comments to the Author**

1. If the authors have adequately addressed your comments raised in a previous round of review and you feel that this manuscript is now acceptable for publication, you may indicate that here to bypass the “Comments to the Author” section, enter your conflict of interest statement in the “Confidential to Editor” section, and submit your "Accept" recommendation.

Reviewer #1: All comments have been addressed

Reviewer #2: All comments have been addressed

Reviewer #3: All comments have been addressed

2. Is the manuscript technically sound, and do the data support the conclusions?

Reviewer #1: Yes

Reviewer #2: No

Reviewer #3: Yes

3. Has the statistical analysis been performed appropriately and rigorously? 

Reviewer #1: Yes

Reviewer #2: Yes

Reviewer #3: Yes

4. Have the authors made all data underlying the findings in their manuscript fully available?

Reviewer #1: Yes

Reviewer #2: Yes

Reviewer #3: Yes

5. Is the manuscript presented in an intelligible fashion and written in standard English?

Reviewer #1: Yes

Reviewer #2: Yes

Reviewer #3: Yes

6. Review Comments to the Author

Reviewer #1: The manuscript has been significantly improved in this revised version. The authors have successfully addressed all the concerns raised during the first round of review. They have provided additional experimental evidence linking the FGF2–PI3K–Akt1 pathway to Lyar expression and have strengthened the mechanistic link between Lyar, Brd2, and adipogenic suppression. The clarification of statistical methods and the tempering of overly speculative claims (such as the aging-related methylation hypothesis) have enhanced the overall quality and credibility of the work.

Reviewer #2: I appreciate the authors’ substantial efforts in revising the manuscript. In particular, the Materials and Methods section has been markedly improved and is now much more detailed, making the study substantially easier to follow and reproduce. The addition of further experimental data, including the Brd2 knockdown experiment and the Akt1-related analysis, has also strengthened several aspects of the manuscript. However, important concerns still remain regarding the strength of the mechanistic interpretation and the alignment between the presented data and the conclusions. Therefore, I consider that the manuscript still requires major revision before it can be considered for acceptance. Below, I summarize my remaining concerns in a point-by-point manner.

1. The Abstract still contains overly strong mechanistic wording.

The revised Abstract still includes statements such as “interacted with Brd2 to suppress adipocyte differentiation,”, “was linked to diminished FGF2–PI3K–Akt1 signaling activity,” and “acts as a context-dependent modulator … balancing proliferation and differentiation.”

These expressions remain too strong relative to the current evidence. The Abstract should more clearly distinguish demonstrated findings from proposed interpretations, using more cautious phrasing such as “may partly mediate,” “was associated with,” or “may act as.”

2. The Introduction still overstates the position of Lyar within the Tak1 signaling framework.

The revised text still presents Lyar in a manner that reads more definitive than the data support. Expressions implying that Lyar is a major regulatory determinant within the TGF-β–Tak1 axis should be toned down. At this stage, the data support Lyar as a candidate or potential contributor, rather than an established key determinant.

3. The interpretation of the Lyar knockdown phenotype remains overstated.

The simultaneous upregulation of early osteogenic and adipogenic markers after Lyar knockdown does not, by itself, justify the conclusion that Lyar suppression promotes lineage commitment. A more cautious interpretation—such as loss of stem-like properties or a more general differentiation-prone state—remains equally plausible. Since later-stage adipogenic readouts under knockdown conditions are still not available, the relevant wording in the Discussion should be revised accordingly. The response acknowledges this limitation, but the corresponding interpretation in the manuscript remains too strong.

4. The claim that Lyar governs the balance between self-renewal and adipogenic commitment remains too strong.

Even with the additional supplemental data, the evidence does not yet justify wording that suggests Lyar “governs the balance” between these two states. The current data support involvement of Lyar in these phenotypes, but not such a definitive regulatory conclusion. More neutral phrasing would be more appropriate. Both the original concern and the authors’ response confirm that this point remains central to the interpretation of Figure 3A.

5. The functional interpretation of the Lyar–Brd2 axis should remain explicitly partial.

The added Brd2 knockdown experiment is valuable and clearly improves the manuscript. However, the evidence still indicates, at most, a partial contribution of Brd2 to the anti-adipogenic effect associated with Lyar, and this conclusion is based mainly on early adipogenic marker changes. Therefore, wording implying that Lyar suppresses adipogenesis through Brd2 remains too strong. This section should instead state that the data suggest that Brd2 may partly mediate the anti-adipogenic effect associated with Lyar.

6. The proposed FGF2–PI3K–Akt1–Lyar pathway is still presented too causally.

The additional siAkt1 experiment strengthens the association between Akt1 signaling and Lyar expression. However, the full upstream pathway is still not established mechanistically. In addition, the response to the earlier concern about upstream regulation does not directly address that issue, and instead shifts to the TAK1–Lyar interaction. Accordingly, the manuscript should present FGF2–PI3K–Akt1 as a plausible upstream pathway associated with Lyar expression, rather than an established causal cascade.

7. The limitation of using GAPDH as the sole qPCR normalizer should be acknowledged more explicitly.

I do not consider this issue fatal to the main descriptive conclusion of the study, because the age-associated reduction of Lyar is also supported by protein/proteomic data. However, several mechanistic interpretations still rely heavily on qPCR-based readouts. In an aging-related mesenchymal system, the stability of GAPDH should not be assumed without discussion. At minimum, this should be acknowledged as a methodological limitation, and the corresponding interpretation should be moderated accordingly.

8. Figure legends remain insufficiently informative.

Although the legends were revised, several still do not contain enough methodological detail for readers to interpret the data independently. In particular, the qPCR normalization gene and the adipogenic induction conditions should be stated explicitly where relevant. This issue was raised previously, and the response indicates revision, but the legends still appear insufficiently detailed.

9. The quantitative support for some image-based conclusions remains limited.

The supplementary quantification added in 3T3-L1 cells is noted, but this does not fully compensate for the limited quantification in the main BMMSC dataset. Where conclusions are drawn from representative staining or immunoblot images, stronger quantitative support would still be preferable. If such quantification is not feasible, the corresponding textual claims should be moderated. The response itself indicates that the BMMSC experiments could not yield reliable ORO quantification, which should be reflected in the strength of the manuscript’s claims.

10. The supplementary proteomic dataset is still not sufficiently reader-friendly.

Although threshold information and highlighted molecules were reportedly added, the supplementary dataset still reads largely as a raw list rather than a clearly guided resource for the reader. Since proteins such as Akt1 and others are important for the interpretation in the main text, a clearer summary table, annotation, or visual guide would improve the connection between the supplementary proteomic data and the mechanistic discussion. The current supplementary tables still appear difficult to navigate in relation to the main argument.

11. The missing or unclear citation for the statement “We previously demonstrated …” should be rechecked carefully.

Although the response letter states that this issue has been corrected, I still could not clearly confirm the appropriate citation in the revised manuscript for the sentence stating that the authors “previously demonstrated that Tak1 plays a pivotal role in regulating the proliferation and differentiation of BMMSCs.” Please recheck this sentence carefully and ensure that the corresponding reference is explicitly provided at the end of the statement. This is a relatively small point compared with the interpretative issues above, but it should still be corrected for completeness.

Reviewer #3: The authors have adequately addressed comments raised in a previous round of review and I feel that this manuscript is now acceptable for publication.

7. PLOS authors have the option to publish the peer review history of their article (what does this mean?). If published, this will include your full peer review and any attached files.

Reviewer #1: No

Reviewer #2: No

Reviewer #3: No

---

## [Author Response · Author response to Decision Letter 2]

22 Apr 2026

We would like to express our sincere gratitude for your careful and thorough re-evaluation of our manuscript.

We greatly appreciate your insightful and constructive comments, all of which we consider to be well-founded and highly valuable.

We have carefully addressed each of the points raised by Reviewer 2, as detailed below.

Q1–6. We have revised all statements that were overly definitive, potentially overestimated, or based on limited data. In particular, the relevant sections in the Discussion have been comprehensively revised. As numerous modifications were made, we would be grateful if you could refer to the revised text highlighted in red.

Q7. We have addressed this point by adding a new section describing the limitations of the study.

Q8. We have revised the figure legends accordingly. In addition, the adipogenic differentiation conditions have now been clearly described in the Materials and Methods section.

Q9. We have added the following statement to the relevant section of the Discussion:

Although the extent to which these mechanisms operate in primary BMMSCs remains to be determined, similar effects were observed in 3T3-L1 cells (Supplementary Figure 2). However, additional quantification in 3T3-L1 cells does not fully compensate for the limited quantitative support in primary BMMSCs, and therefore conclusions based on representative images should be interpreted with caution.

Q10. We have revised the Supplemental file to improve clarity and facilitate the reader’s understanding, including the addition of highlights.

Q11. We sincerely apologize for this omission and have now added the appropriate citation.

---

## [Decision Letter · Decision Letter 2]

6 May 2026

Decreased expression of Ly-1 antibody reactive clone (Lyar) triggers enhanced adipogenesis of bone marrow mesenchymal stromal cells in aged bone marrow

PONE-D-25-42314R2

Dear Dr. Teramura,

We’re pleased to inform you that your manuscript has been judged scientifically suitable for publication and will be formally accepted for publication once it meets all outstanding technical requirements.

Kind regards,

Sadiq Umar

Academic Editor

PLOS One

---

## [Editor Report · Acceptance letter]

PONE-D-25-42314R2

PLOS One

Dear Dr. Teramura,

I'm pleased to inform you that your manuscript has been deemed suitable for publication in PLOS One. Congratulations! Your manuscript is now being handed over to our production team.

Kind regards,

on behalf of

Dr. Sadiq Umar

Academic Editor

PLOS One